**METHOD**

# Contiguous and stochastic CHH methylation patterns of plant DRM2 and CMT2 revealed by single-read methylome analysis

Keith D. Harris and Assaf Zemach*

*Correspondence:
assafze@tauex.tau.ac.il
School of Plant Sciences and Food
Security, Tel Aviv University, Haim
Levanon, Tel Aviv, Israel

## Abstract

Cytosine methylome data is commonly generated through next-generation sequencing, with analyses averaging methylation states of individual reads. We propose an alternative method of analysing single-read methylome data. Using this method, we identify patterns relating to the mechanism of two plant non-CG-methylating enzymes, CMT2 and DRM2. CMT2-methylated regions show higher stochasticity, while DRM2-methylated regions have higher variation among cells. Based on these patterns, we develop a classifier that predicts enzyme activity in different species and tissues. To facilitate further single-read analyses, we develop a genome browser, SRBrowse, optimised for visualising and analysing sequencing data at single-read resolution.

**Keywords:** DNA methylation, Epigenetic variation, NGS analyses, Genome browser

## Background

DNA methylation is a conserved epigenetic mechanism that regulates genome stability and expression in diverse eukaryotes [1–4]. This regulation is based on a dynamic addition or removal of a methyl group to/from the fifth carbon of a cytosine residue. DNA methylation appears in distinct genomic features, such as genes and transposable elements (TEs), and in different chromatin states, such as heterochromatin and euchromatin [2, 5–9]. In plants, DNA methylation occurs in three contexts: CG, CHG and CHH (where H is any base except G). These contexts are differentially regulated by four DNA methyltransferase (DNMT) families that share a conserved methyl-transferase domain (MTD). METHYLTRANSFERASE1 (MET1) recognises hemi-methylated CG following DNA replication and methylates the naked cytosine in the daughter strand [10, 11]. CHROMOMETHYLASEs (CMTs), which are plant-specific DNMTs, bind histone H3 lysine 9 (H3K9me2) heterochromatin via a chromodomain (CD) to methylate non-CG

contexts [12]. In flowering plants, CMT3 methylates mostly CHG sites, whereas CMT2 methylates mostly CHH sites [13, 14]. The CHH methylation state is additionally regulated by plant DNMT3 orthologues or homologs, i.e. the DOMAINS REARRANGED METHYLASEs (DRMs) [15, 16]. Similar to animal DNMT3, plant DNMT3 and DRMs function as de novo methylases, establishing methylation on unmethylated sites.

The relationship between changes in DNA methylation patterns and gene expression is not trivial, as it involves a non-linear, additive effect of multiple methylation contexts, along with the effect of additional levels of epigenetic regulation, including chromatin structure, and histone position and modifications [1, 17, 18]. Additionally, the most common method for studying DNA methylation (bisulfite sequencing, or BS-seq) does not provide information on the methylation states of individual cells. BS-seq involves a chemical reaction that converts unmethylated cytosines into uracil, which are subsequently read as thymine when sequenced [19]. Sequencing data produced by BS-seq consists of short DNA fragments originating from a random subset of cells in the sample tissue; cytosines that have not been converted to thymine are assumed to represent methylated cytosines in the source genome [20, 21]. Hypothetically, each read relates to the methylation state of a single cell in the sample, and so the collection of reads will reflect methylation heterogeneity present in the sample tissue. However, BS-seq methylome data is most commonly averaged among reads overlapping the same region. Thus, the output signal of BS-seq analysis pipelines combines populations that may have fundamentally different methylation levels.

While there are alternative methods for generating methylomes that address this issue, namely single-cell BS-seq, these methods are currently not feasible for all organisms and tissues [22–25]. As a consequence, most currently available methylome data is not single cell; analyses that can decode additional dimensions of information from this type of data are of high potential value in producing more insights from new and existing data. One such analysis was recently proposed to produce a heterogeneity signal from CG methylation patterns among cells [26]. This tool calculates the Shannon entropy of reads overlapping a set of CG sites and identifies unique patterns of methylation within this subsample, as relating to heterogeneity within the sample population of cells. Similarly, a method has been proposed for identifying subtypes of cells within heterogeneous BS-seq samples, by observing differential regulation of CG methylation among reads [27].

We were interested in extracting additional information from BS-seq data, specifically relating to CHH methylation, which could identify patterns of methylation associated with specific genomic regions, chromatin structure or methylase activity. To this end, we designed a single-read analysis pipeline that extrapolates multiple dimensions of methylation variation, using NGS reads either from a single region (collection of CHH sites) or from functionally similar sets of regions. This analysis revealed that DRM2 and CMT2 have distinctive methylation patterns at both single-cell and population levels. CMT2-methylated reads and regions are more stochastically methylated than DRM2-methylated reads. These findings make new predictions regarding the distinct mechanisms of CHH-methylating enzymes. By characterising these patterns in *Arabidopsis thaliana* mutants of these enzymes, we developed a classifier that can predict the identity of the enzyme that methylates a particular region. Importantly, the classifier does not rely on a comparison to mutants of the same species or tissue. At a genome level, it can predict the presence or absence of DRM2-like or CMT2-like activity. After validating the classifier, we used it

to predict null DRM2 CHH methylation activity and to associate the CMT2 methylation pattern to that of the DNMT3 methylation signal in early land plant species and human cells.

Our analyses use BS-seq data at single-read resolution. To facilitate further analyses, we developed a genome browser, "Single-Read Browser" (SRBrowse), that is optimised for visualising and analysing NGS data at single-read resolution. The tool, which has a unified user interface for browsing and analyses, can directly process local NGS data or NCBI accessions into an optimised format for display in the genome browser.

## Results

### Designing a single-read analysis pipeline for CHH methylation

Single-read analyses can be usefully applied to any NGS data where short reads vary in a way that reflects biological variation. With BS-seq data, the importance of analyses at this resolution is that methylation varies between cells meaningfully, with each read hypothetically reflecting the state of a single cell. We chose to focus on CHH methylation for a number of reasons: (i) CHH sites are 2–3 times more common than CG/CHG sites. Given that site density dictates the amount of information that can be deduced from a single read, contexts with a higher density allow more data to be retrieved from individual samples with low coverage. (ii) As opposed to CG sites, the methylation of which is mostly binary, CHH sites are mostly partially methylated [20, 21, 28] (sites that are either unmethylated or fully methylated have low or zero variation among reads). (iii) CHH methylation is known to vary between tissues [29]; in itself, the fact that CHH sites are partially methylated suggests that most CHH sites are differentially methylated between cells of the sample tissue [20, 21]. (iv) CHH sites are methylated by two types of DNMTs, DRM and CMT, the activity of which is regulated by distinct molecular mechanisms, RdDM- and DDM1-dependent respectively [13, 14]. Thus, focusing on CHH sites might expose the potential variation between regions of the same sample due to the different mechanisms involved.

There are a number of factors that limit the maximal region size used to compare reads, mainly (1) the average read length and coverage of the sequencing library and (2) the frequency of the specific methylated context. The expected number of reads per region for a given library can be calculated as:

$$\text{expected reads per region} = \text{coverage} \cdot \left( 1 - \frac{\text{region size}}{\text{read length}} \right)$$

This relationship is illustrated in Additional file 1: Figure S1a. The frequency of the methylation context is also important to consider, as selecting regions rich in a particular context can limit analyses to a small subset of the data and thus bias the results of the analysis (Additional file 1: Fig. S1b). Different types of analyses can utilise different filter options (e.g. depending on the characteristics of the region of interest). For all analyses except where noted otherwise, regions were selected with 5 CHH sites, up to 30 bp length. In a wild type *A. thaliana* sample [14], this includes 58% of regions from TEs containing 5 CHH sites with ≥ 5% methylation (Additional file 1: Fig. S1c).

In order to study individual- and population-level variation of methylation, the pipeline segments the genome into short regions of a limited length of similar functional elements or annotations, e.g. TEs, genes, exons, histone marks and chromatin structure. Due to the

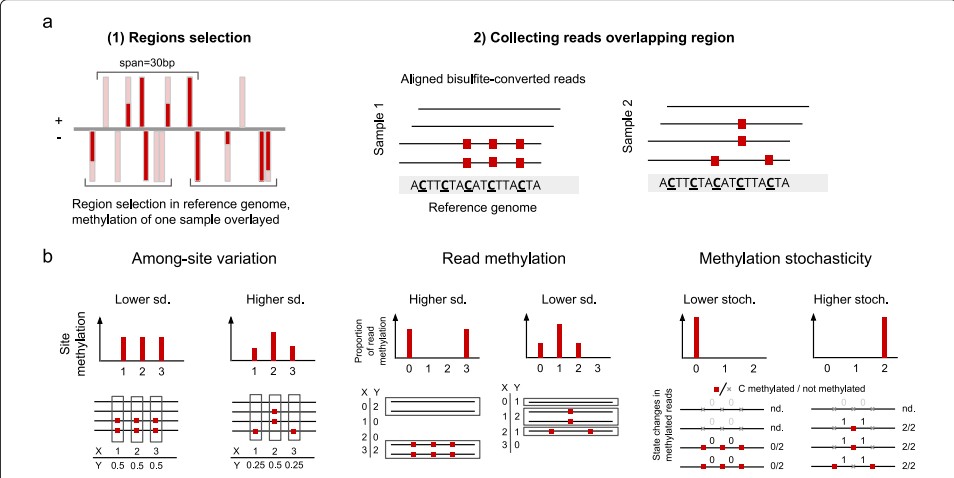

**Fig. 1** Pipeline for analysing variation of single-read data regarding CHH-methylated regions. **a** Schematic representation of pipeline: regions of up to a set length in base pairs with set number of CHH sites are selected; reads overlapping these sites are then quantified by the number of methylated CHH sites. **b** Schematic representation of different types of variation within methylated regions: (i) among-site variation, relating to differences in methylation average between adjacent CHH sites; (ii) read methylation variation, relating to differences in methylation level among reads overlapping the same region, taken to represent the sample population of cells; and (iii) stochasticity, relating to differences between adjacent CHH sites within each read

asymmetry of CHH sites, these regions are defined per strand. Here we used regions spanning a maximum of 30 base pairs with 5 CHH sites (Fig. 1a). Reads overlapping all 5 CHH sites are then scored according to the methylation state of these sites, within the read (Fig. 1a). We defined three features of methylation variation: (1) the standard deviation of CHH site methylation, (2) the standard deviation of read methylation and (3) stochasticity (Fig. 1b). These features are potentially related to functional differences between methylation patterns: (1) higher variation among sites can reflect fluctuations in the methylation signal and/or CHH subcontext specificity of the enzyme; (2) higher variation among reads can reflect differential regulation of methylation among cells composing the sample; and (3) higher stochasticity could reflect subcontext specificity.

### Read-level CMT2 CHH methylation activity is more stochastic than that of DRM2

To compare CMT2 and DRM2 methylation patterns, we used BS-seq data of mutants defective in the respective enzymes [14]. The mutants included *cmt2* (three mutated alleles) and *drd1* and *drm2* (which both affect DRM2 activity [14, 30]). The assumption was that, given the complementary activity of these enzymes, the remaining CHH methylation in *cmt2* should consist of DRM2-methylated reads, whereas methylation in *drd1* or *drm2* should consist of CMT2-methylated reads [13, 14]. As the methylation pattern in the three *cmt2* mutants was similar, we combined the methylation data of these samples.

Initially, we tested whether the methylation level of individual reads differs between the mutants on a genomic scale. To do this, reads were collected from regions conforming to the standard parameters of our analysis (30-bp max region span, 5 CHH sites). Reads were scored based on the methylation of the CHH sites from 0 to 5.

Figure 2a shows the distribution of read methylation level resulting from the above analysis in each of the mutants, selected from regions methylated at three different minimal levels (5%, 20% and 40%). Methylated reads from *drd1* and *drm2* show a different

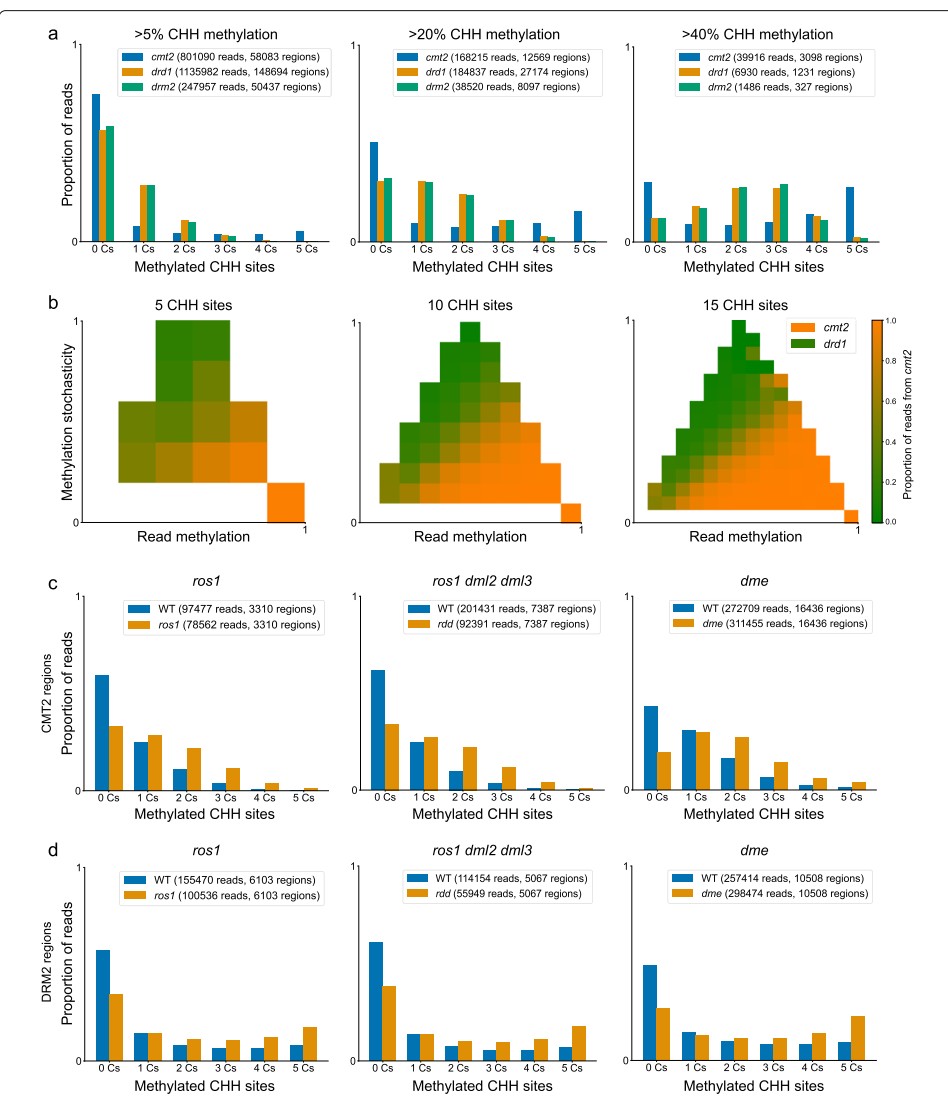

**Fig. 2** Single-read analysis of CHH methylation in CHH methylase and demethylase mutants. **a** Methylation states of reads from CHH methylation mutants. Binned methylation states of all reads overlapping regions matching the criteria (5 CHH sites, maximum 30 bp length) are plotted, demonstrating differences in methylation intensity between *cmt2*, *drd1* and *drm2*. Each panel presents regions filtered according to the averaged region methylation level, as indicated above each panel. The methylation state is defined, as in Fig. 1, as the number of CHH sites in the region that are methylated in each read. **b** Analysis of methylation patterns within reads. Reads were selected from regions containing 5, 10 and 15 CHH sites (as indicated above the panels) and with a maximum length of 100 bp and binned according to read methylation and methylation stochasticity (as defined in Fig. 1b). White areas indicate no data. **c**, **d** Comparison of demethylase mutant read methylation and respective wild type samples: **c** sites methylated in the *drd1* mutant (i.e. CMT2-methylated sites) that are hypermethylated in the demethylase mutants; **d** sites methylated in the *cmt2* mutant (i.e. DRM2-methylated sites) that are hypermethylated in the demethylase mutants. To select CMT2- and DRM2-methylated sites in the *ros1-4* and *rdd* mutants, the *drd1* and *cmt2* mutants from [14] were used, respectively. For *dme*, *drm1/2* and *cmt2* mutants from [31] were used

distribution from the combined *cmt2* data: while *cmt2* retains a proportion of fully methylated reads, *drd1* and *drm2* retain mainly partially methylated reads, with a low proportion of fully methylated reads. This pattern is present even in regions with high (≥ 40%) average methylation (Fig. 2a, rightmost panel). The difference between lowly and highly methylated *drd1* or *drm2* regions is explained exclusively by the methylation state

of partially methylated reads (Additional file 1: Fig. S2a-b). In comparison, the proportion of partially methylated reads between lowly and highly methylated regions in *cmt2* is similar, while methylation is correlated to the proportion of fully methylated reads (Additional file 1: Fig. S2a). Due to the relatively low coverage of the *drm2* mutant, we used *drd1* for the subsequent analyses, but validated the patterns identified in *drd1* using the *drm2* mutant and *drm2* mutants from other studies.

Figure 2a demonstrates that reads from *drd1* and *cmt2* have different methylation levels. An additional dimension of variation among reads is the stochasticity of methylation. We defined this as the distribution of methylation within reads and quantified it by counting the number of changes in methylation within the read (e.g. a methylated CHH site adjacent to an unmethylated CHH site on the same strand) out of the total possible number of changes (illustrated in Fig. 1b). Figure 2b demonstrates the separation between reads from the respective mutants according to their methylation level and stochasticity: while *drd1* has reads with lower methylation and higher stochasticity, *cmt2* has reads with higher methylation and lower stochasticity. This correlation persists in regions with different CHH content, as shown, 5–15 sites per region (Fig. 2b), suggesting that this pattern does not depend on the density of CHH sites. This result is also consistent for the mutant alleles composing the *cmt2* sample (i.e. *cmt2-4*, *cmt2-5* and *cmt2-6*) and *drm2* (Additional file 1: Fig. S2c). Overall, these results suggest that CMT2 is associated with a CHH methylation pattern that is more stochastic than that associated with DRM2.

### CMT2 and DRM2 CHH methylation patterns are not dependent on demethylase activity

*A. thaliana* DNA demethylases regulate DNA methylation levels through direct removal of methylated cytosine bases from all cytosine sequence contexts [20, 32–34]. Therefore, distinct patterns of CHH methylation in the DNMT mutants could result from demethylase activity. To test this hypothetical scenario, we analysed three different demethylase mutants: single mutants repressor of silencing (*ros1-4*), demeter (*dme-2*) and the triple mutant *ros1-3*, demeter-like protein 2 (*dml2-1*) and *dml3-1* (*rdd*). Regions methylated in either *drd1* or *cmt2* were analysed as representing regions methylated by the complementary enzyme; the distribution of read methylation at these regions for each of the wild type/demethylase mutant pairs was plotted.

Figure 2c and d summarises this comparison. Figure 2c presents data from CMT2-methylated regions (methylated in *drd1*), while Fig. 2d presents reads from DRM2-methylated regions (methylated in *cmt2*). For *dme*, *drm2* and *cmt2* mutants of vegetative nucleus tissue from [31] were used. For each demethylase, regions were also selected according to hypermethylation (> 10% increase in methylation) relative to the respective wild type sample. The enzyme-associated pattern is present in the demethylase mutants and its respective wild type sample (Fig. 2c, d): in CMT2 regions, both the mutant and wild type have a low proportion of fully methylated reads, with hypermethylation in the mutant correlating to the increase in partially methylated reads (Additional file 1: Fig. S3a-c, left panels); in DRM2 regions, both the mutant and wild type have fully methylated reads, with hypermethylation in the mutant correlating to the increase in fully methylated reads (Additional file 1: Fig. S3a-c, right panels). This suggests that the patterns identified in the CHH methylation mutants are present prior to demethylase activity.

### Variation of CHH methylation among adjacent sites and overlapping reads distinguishes between CMT2 and DRM2 target regions

Analyses of individual reads can reflect the activity of different CHH-methylating enzymes, but the predictive confidence in distinguishing between methylated reads is limited, given that most reads are lowly methylated with a limited range of stochasticity (Additional file 1: Fig. S2c). Hence, to characterise region methylation, we produced methylation features per-region from the single-read data (Fig. 1b). The separation of all methylated regions ($\geq$ 10% average methylation) from the mutant samples is shown in Fig. 3a–c. Paired with the distributions of features of the actual data (solid lines) are features of read datasets generated using a Poisson model of single-C sites, where the chance of methylation per-site, per-read is equal to the mean region methylation (dashed lines). The similarity between the actual and generated data can demonstrate the degree to which the feature distributions of each mutant are explained by stochastic variation at the level of individual CHH sites in reads.

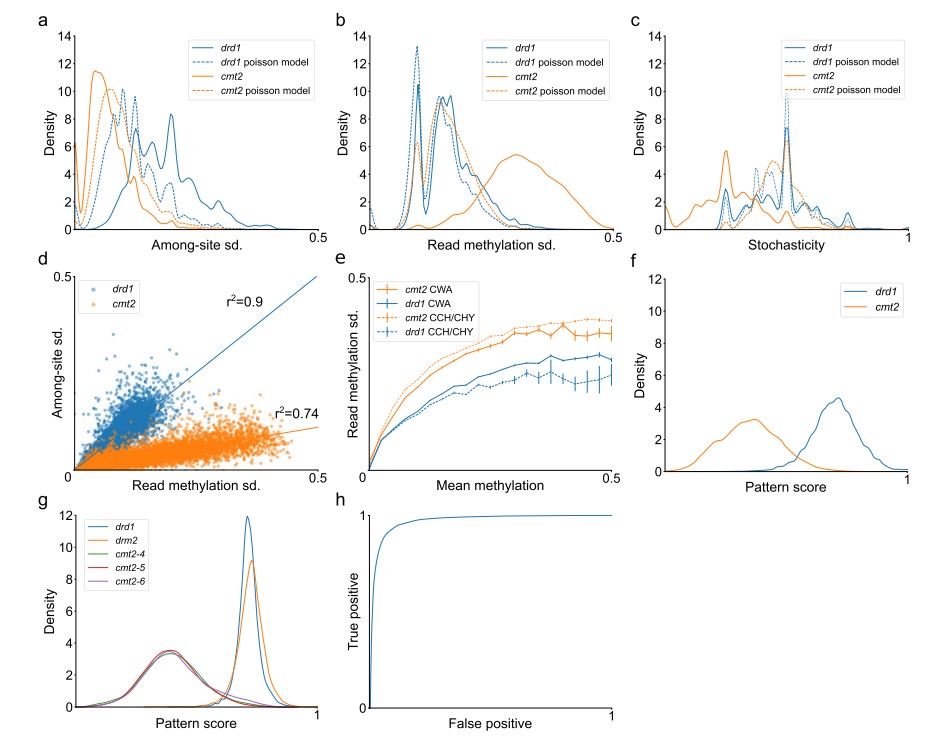

**Fig. 3** Separation of CHH-methylated sites according to methylation variation. **a**–**c** K-density estimate plots of each type of variation in CHH methylation mutants. Distribution of sample data is shown in solid lines, dashed lines show the methylation features of a random Poisson model based on the average methylation of each sample. **d** Separation of whole transposable elements (TEs) according to among-site and read methylation variation in CHH methylation mutants (no minimal methylation level for regions or TEs). Linear regressions for each mutant are drawn as solid lines matching the color of the scatter plot, along with squared correlation coefficients. $p$ value $< 1 \times 10^{-10}$ for both regressions. **e** Differences between CWA and non-CWA subcontexts (CCH/CHY, where Y is C or T) in CHH methylation mutants. Lines represent the average read methylation sd. of a region for a given methylation level. **f** Resulting separation of regions from CHH methylation mutants based on the *pattern score* feature. **g** Distribution of whole TEs according to pattern score in CHH methylation mutants (*cmt2* separated into its composing samples). As opposed to **d**, in **f** and **g**, a minimal region methylation of 10% was used to filter regions. **h** Classifier receiver operating characteristic curve demonstrating separation between regions of CHH methylation mutants as shown in **f**

CMT2-methylated regions have higher variation among sites, lower variation of read methylation level and higher average stochasticity (as suggested by Fig. 2b). Read methylation variation and average read stochasticity from CMT2-methylated regions overlap with the distributions of generated data (Fig. 3a–c), suggesting that, in the *drd1* sample, variation of CMT2-mediated CHH methylation activity among reads is mainly stochastic. By contrast, DRM2-methylated regions have lower variation among sites, higher variation among reads and lower stochasticity (Fig. 3a–c). Variation among reads in DRM2-methylated regions is not stochastic, suggesting that DRM2-mediated CHH methylation in these regions is differentially regulated. Of these factors, variation among reads best predicts the methylating enzyme of the region (Additional file 1: Table S1).

To understand the relationship between these methylation features in the mutant samples, features were analysed at the level of whole functional elements (in this case, TEs), by averaging the features of individual regions contained within each element. This reduces noise caused by low coverage of individual regions. Only TEs with at least two regions with the required minimal coverage and methylation ($\geq$ 10%) are plotted. Read methylation variation and among-site variation are plotted for all TEs (Fig. 3d) and for specific TE superfamilies (Additional file 1: Fig. S4a). In the *drd1* mutant, these features are correlated with a slope of 1, with among-site variation increasing linearly with read methylation variation (Fig. 3d). On the other hand, in the *cmt2* mutant, the two features are correlated with a smaller slope (0.21), with most TEs having a low average among-site variation (Fig. 3d). This was consistent across different TE superfamilies (Additional file 1: Fig. S4a).

CMT2 shows specificity for the CHH subcontext CWA (i.e. CTA or CAA) [20, 21, 35]. As this could contribute to higher variation when analysing all CHH subcontexts, CWA and non-CWA subcontexts were analysed separately. For this comparison, a larger region size of 50 bp was used, given the lower density of CWA sites (4–5 times lower than that of CHH). CWA contexts had higher variation among reads in *drd1*, whereas in *cmt2* these levels are comparable to all CHH subcontexts (Fig. 3e, Additional file 1: Fig. S4a). Among-site variation remains similar. The increase in read methylation variation can be explained by the higher methylation of CWA-methylated regions. In addition, CWA-methylated regions in *drd1* have higher read methylation variation relative to regions methylated at the same level in *drd1*, but still lower than in *cmt2* (Additional file 1: Fig. S4b). In non-CWA and all CHH-sites *drd1* read methylation variation is similar to that of a stochastic model. Methylated reads from CWA-methylated regions show a similar pattern in terms of methylation level and stochasticity, as opposed to non-CWA-methylated regions (Additional file 1: Fig. S4c-d). This suggests that the CMT2 methylation pattern observed in CHH-methylated sites in *drd1* is composed of two distinct patterns; however, even when including only sites for which CMT2 shows specificity, the *drd1* mutant shows higher stochasticity compared to *cmt2* (Additional file 1: Fig. S4b, left panel).

Based on ANOVA of the methylation features in the CHH methylation mutants, we designed a classifier to score regions and whole functional elements in terms of the CHH methylation pattern. The results of the model are presented in Additional file 1: Table S1. The separation of the mutants used to define the classifier based on the *pattern score* feature is presented in Fig. 3f (regions) and Fig. 3e (whole elements), along with the receiver operating characteristic (ROC) curve of region prediction (Fig. 3h). The pattern score feature ranges from 0 to 1, with lower values indicating patterns associated with DRM2 methylation, and higher values indicating patterns associated with CMT2 methylation.

Each mutant shows a single peak of pattern score, and these peaks are aligned for mutants affecting the same enzyme (Fig. 3g). The separation of the mutant samples in Fig. 3g and the ROC curve of the classifier demonstrate the potential of using the classifier to predict enzyme identity. *A. thaliana cmt2* mutants from previous studies and *drm2*-related mutants from multiple species show similar distributions of pattern score, suggesting that this distribution is not specific to the mutant samples used to construct the classifier (Additional file 1: Fig. S4b-c).

### DRM2 and CMT2 methylation patterns are distinct from chromatin structure-dependent patterns

DRM2 and CMT2 function at distinct chromatin environments; DRM2 via RdDM is targeted mainly to euchromatic TEs, whereas CMT2 via H3K9me2 is targeted preferentially to heterochromatic TEs [13, 14, 30]. Accordingly, it is possible that the distinct CHH methylation activities of DRM2 and CMT2 are influenced by the genomic chromatin environment rather by their intrinsic enzymatic activity. In order to test the role of the genomic environment on DRM2 and CMT2 methylation patterns, we correlated pattern score with GC content (a prominent indicator for chromatin structure [14, 18, 36]), for individual CHH-methylated regions in *drd1* and *cmt2* mutants (Fig. 4a). While regions methylated in *drd1* show on average higher GC content than regions methylated in *cmt2*, no correlation was found between GC content and pattern score in either mutant (Fig. 4a). We also correlated pattern score in *cmt2*, *drd1* and *drm2* with the following hetero- and eu-chromatic histone marks, H3K9me2 and H3K4me3, respectively [37]. As with GC content, the mutants are separated by both histone marks and pattern score, but there is no correlation between histone marks and pattern score within each mutant (Fig. 4b, c). These results suggest that the methylation patterns associated with DRM2 and CMT2 are not derived from differences between chromatin environments.

### Plant and human DNMT3s show similar CHH methylation patterns to that of angiosperm CMT2

By identifying methylation patterns associated with either DRM2 or CMT2, the classifier can predict the presence or absence of the activity of either enzyme in samples from

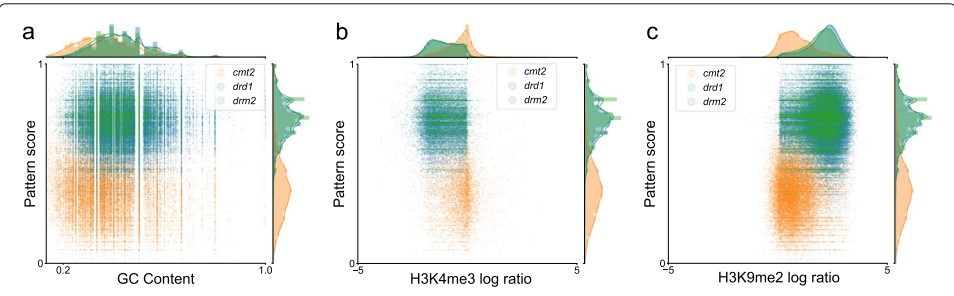

**Fig. 4** DRM2 and CMT2 CHH methylation patterns are distinct from chromatin structure-dependent patterns. Correlation between chromatin features and pattern score, in CHH-methylated regions from *cmt2*, *drd1* and *drm2*: **a** GC content and pattern score (*cmt2* $r^2 < 1 \times 10^2$, $p$ value $< 1 \times 10^{-10}$; *drd1* $r^2 < 1 \times 10^{-3}$, $p$ value $< 1 \times 10^{10}$; *drm2* $r^2 < 1 \times 10^{-2}$, $p$ value $< 1 \times 10^{10}$). Due to the requirement of 5 CHH sites, GC content is above a minimal value ($> 0.133$); **b** H3K4me3 log fold difference and pattern score (*cmt2* $r^2 < 1 \times 10^{-2}$, $p$ value $< 1 \times 10^{-8}$; *drd1* $r^2 < 1 \times 10^{-2}$, $p$ value $= 0.15$; *drm2* $r^2 < 1 \times 10^{-2}$, $p$ value $< 1 \times 10^{-5}$); **c** H3K9me2 log fold difference and pattern score (*cmt2* $r^2 = 0.02$, $p$ value $< 1 \times 10^{-10}$; *drd1* $r^2 < 1 \times 10^{-2}$, $p$ value $< 1 \times 10^{-3}$; *drm2* $r^2 < 1 \times 10^{-2}$, $p$ value $< 1 \times 10^{-8}$)

different tissues and species. Currently, in the absence of mutants showing partially reduced CHH methylation, it is unclear whether the total methylation pattern present in a given sample is derived from one or more CHH-methylating enzymes, and this distinction cannot be made based on average methylation alone.

Mutants that possess only one active CHH-methylating enzyme present a simplified case for the classifier, given that there is only one pattern. To assess the ability of the classifier to identify distinct patterns in more diverse samples, the classifier was applied to wild type samples of multiple species (Fig. 5a). Of the species analysed, *A. thaliana*, *Oryza sativa* and *Solanum lycopersicum* had two peaks, while *Physcomitrella patens* and *Marchantia polymorpha* had only one peak. In *A. thaliana*, the two peaks associate with either of the CHH methylation mutants shown in Fig. 3g. *S. lycopersicum* and *O. sativa* have two peaks, similarly to *A. thaliana*, which are also aligned to the *A. thaliana* CHH methylation mutants; however, the ratio between the peaks is different. This ratio relates to the frequency of TEs regulated by either CMT2 or DRM2. For example, rice is known for its exceptional number of MITEs (130k) targeted by RNA-directed DNA methylation (RdDM) and DRMs [38].

Both *P. patens* and *M. polymorpha* have a single peak that is associated with the pattern score of the *A. thaliana drd1* mutant (Fig. 5a). In addition, reads from these species have high stochasticity, similar to that of the *drd1* mutant (Additional file 1: Fig. S6). *P. patens* has one dominant CHH methylation enzyme, DNMT3, with trivial methylation activity by DRMs [16]. Finding a single pattern distribution in *P. patens* that is similar to that of CMT2 substantiates the trivial CHH methylation by PpDRMs and suggests that the CHH methylation activity of DNMT3 is comparable to that of CMT2. Similarly to *P. patens*, *M. poylmorpha* contains DRM and DNMT3 and is missing CMT2 [40]. The classifier identified a single enzyme peak in the *M. polymorpha* methylome that overlaps that of PpDNMT3 and CMT2 (Fig. 5a), predicting that, similarly to *P. patens*, DNMT3 rather than DRMs are its main CHH methylases. Figure 5b suggests that *P. patens* and *M. polymorpha* read methylation variation is higher than that of *A. thaliana drd1*, but lower than that of *A. thaliana cmt2*. In addition, both *P. patens* and *M. polymorpha* have higher among-site variation compared to *A. thaliana cmt2* (Fig. 5b). However, this difference relates partly to differences in methylation level of these samples: when comparing region features for a given region methylation level, *P. patens*, *M. polymorpha* and *A. thaliana drd1* show more minor differences in among-site and read methylation variation (Fig. 5c, d).

While CG is the predominant methylation context in animals, non-CG methylation can be enhanced in particular tissues, such as the brain [3, 41]. Non-CG methylation in mammals (also called CH methylation) is mediated by DNMT3s. In human, two DNMT3s, DNMT3a and DNMT3b, were found to mediate CH methylation [3]. Thus, to test how many CH methylation patterns exist in human data and their relationship to those found in plants, we next ran our single-read method on human methylomes derived from brain tissue. Applied to human CH data using the same parameters as for CHH analyses, our classifier detected a single peak of pattern score that overlaps that of plant DNMT3 and CMT2 enzymes (Fig. 5e). Distributions of variations of among-site and read methylation also show only a single peak of activity (Fig. 5f). These results suggest that CH methylation in neurons has a single dominant pattern that is similar to that of plant DNMT3 and CMT2.

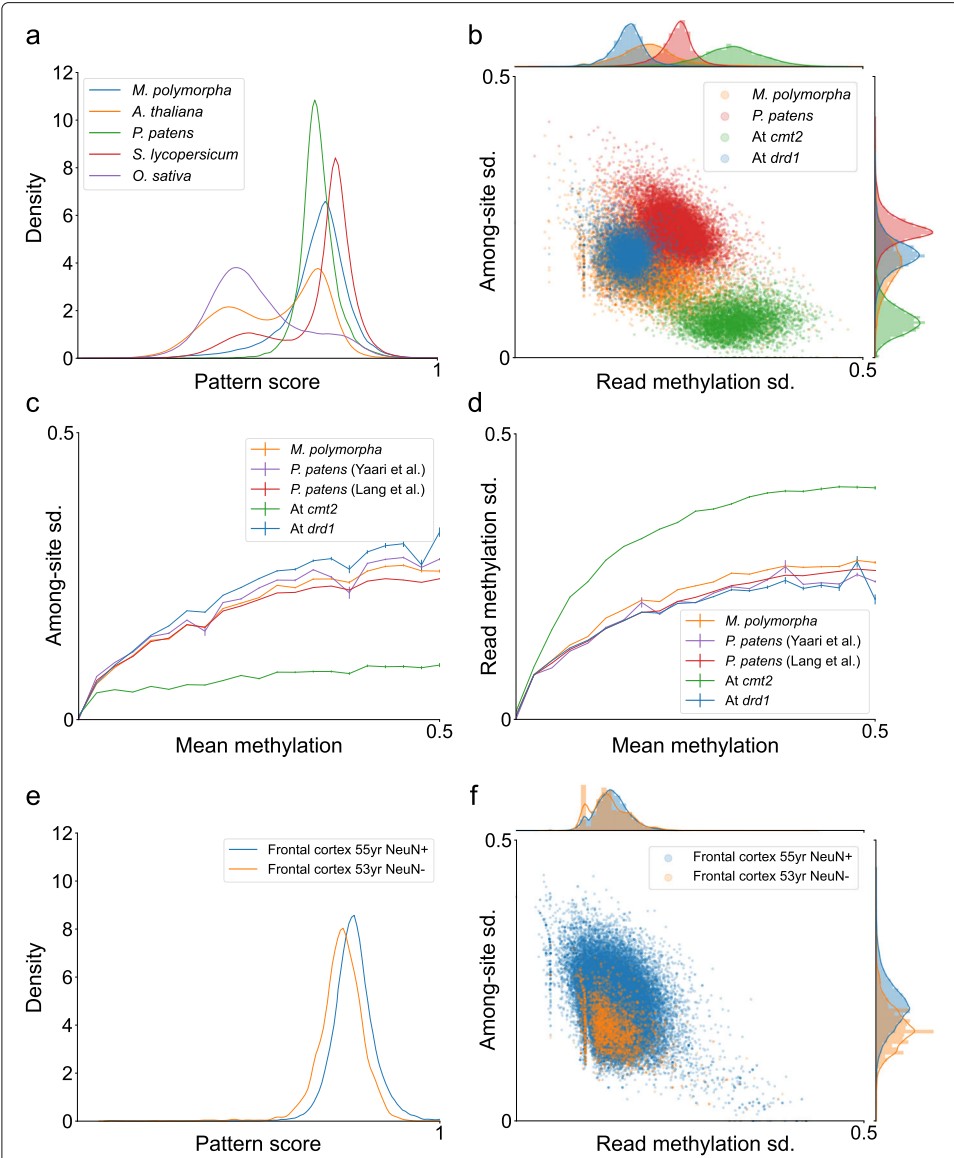

**Fig. 5** Identification of methylation patterns associated with CMT2 and DRM2 in multiple species. **a** Distribution of TEs in multiple species according to pattern score. Single peaks indicate the identification of only one pattern signal; two peaks indicate the presence of two pattern signals. **b** TE methylation variation features in *P. patens* and *M. polymorpha* that lack a peak aligned with DRM2-associated patterns, alongside CHH methylation mutants from *A. thaliana* (denoted "At"). **c**, **d** Differences of region among-site sd. and read methylation sd. among *M. polymorpha*, *P. patens* and CHH methylation mutants of *A. thaliana* (denoted "At"), with respect to region methylation: lines represent the average among-site sd. **c** or read methylation sd. **d** of a region for a given methylation level. All regions are plotted (i.e. without a minimal methylation level). **e** Analysis of the pattern score of exon CH methylation in human samples from the frontal cortex [39]. **f** Methylation variation features of individual exons in human samples from **e**

Conclusively, these results demonstrate the use of the pattern classifier in predicting the presence or absence of CMT2- or DRM2-like methylating activity at a genomic scale.

### Tissue-specific samples have different proportions of CMT2/DRM2-methylated regions

The pattern classifier relies on methylation features the range of which may be biased by sample composition. For example, read methylation may vary less within homogeneous

samples, if methylation patterns are similar between cells. Given that read methylation variation is the strongest predictor of enzyme identity ($r^2 = 0.589$), the effectiveness of the classifier may be limited in such samples.

In order to assess the ability of the classifier to function in tissue-specific samples, we used datasets from two studies that produced methylomes of sperm and vegetative nucleus cells [31], and root tissue subsamples [29]. Given that, in each of these studies, altered regulation of CMT2/DRM2 activity was observed in one or more of the samples, this analysis also provided a means of validating the predictions of the classifier.

Figure 6a and b demonstrate the differences in CHH methylation regulation among tissue-specific samples of *A. thaliana*: all samples contain two peaks; however, in some tissues, enzyme activity shifts, with both sperm and root tip having more DRM2-methylated TEs. *A. thaliana drm1/2* and *cmt2* mutants from the same study [31] were also analysed, and both vegetative nucleus and sperm mutants are distinguishable based on the pattern score analysis (Fig. 6c, d). As noted above, the wild type sperm sample has less CMT2 activity compared to the vegetative nucleus and other wild type *A. thaliana* samples analysed, confirming previous findings showing reduced CHH methylation in heterochromatic TEs targeted by CMT2 [31].

Individual TEs and regions that are methylated in both sperm and vegetative nucleus are on average more similar to DRM2-methylated regions (Additional file 1: Fig. S7a-b). In addition, relative to other DRM2-related mutants, the pattern score peak of the sperm *drm1/2* mutant is more DRM2-like (Fig. 6d). Interestingly, sperm *drm1/2* has higher read methylation variation, but lower among-site variation (Fig. 6b). This reflects the signal in the wild type sperm sample, in which overall among-site variation is low compared to other *A. thaliana* wild type tissues (Fig. 6b).

Compared to wild type sperm, *drm1/2* sperm TEs are more regulated by CMT2 (Fig. 6f). This change is partly due to the loss of DRM2-methylated regions from TEs, rather than CMT2 methylating previously DRM2-methylated regions. However, the same change is observed also when comparing individual regions: regions retaining methylation in *drm1/2* sperm, in particular regions that are defined by the classifier as regulated by DRM2 in the wild type sample, are shifted towards a CMT2-like signal (Additional file 1: Fig. S7c, left panel). In the vegetative nucleus, the same shift is observed (Fig. 6f; Additional file 1: Fig. S7d, left panel). In the *cmt2* mutant, in both sperm and vegetative nucleus, no change is observed at the level of individual regions (Additional file 1: Fig. S7c-d, right panels).

We also analysed three root samples that differed in their pattern score distributions for whole TEs: root tip (RT), columella root cap (CRC) and lower columella (LC) [29]. The RT sample contains both CRC and LC, but is in itself different from other wild type tissues analysed (Fig. 6a). Similarly to sperm, it has a lower average among-site variation. In terms of pattern score, all three samples have two peaks (Additional file 1: Fig. S7e), but the distributions do not overlap. A comparison of pattern score of individual TEs between samples shows that in the LC sample, all TEs are shifted towards a DRM2-like signal (Additional file 1: Fig. S7f); TEs with intermediate signals (e.g. regulated by both enzymes) are more shifted than those with more defined DRM2/CMT2 signals (e.g. regulated by one enzyme). The same shift is present also in the CRC sample relative to RT (Additional file 1: Fig. S7g). Overall, these results demonstrate the ability of the classifier to predict changes in the activity level of DRM2 and CMT2 in different tissues.

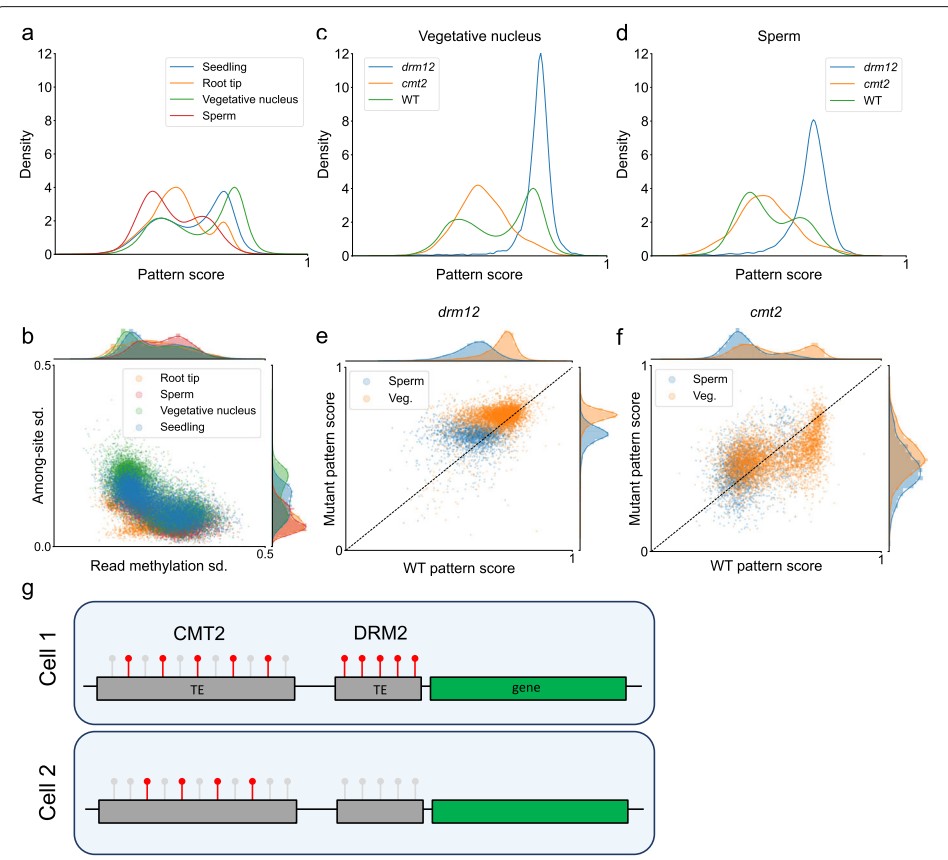

**Fig. 6** Analysis of tissue-specific methylation patterns. **a** Distribution of TEs according to pattern score in *A. thaliana* samples from seedling and specific tissues. **b** Separation of TEs in *A. thaliana* wild type tissues according to read and among-site CHH methylation variation. **c**, **d** Distribution of TEs according to pattern score in *A. thaliana* vegetative nucleus (**c**) and sperm (**d**) wild type samples, alongside *drm1/2* and *cmt2* mutants of the respective tissues. **e**, **f** Correlation between pattern score of individual TEs in wild type compared to mutant tissues. Only elements that were methylated in both wild type and respective mutant were selected. Each dot represents a single TE, with the distance from dashed line ($f(x) = x$) indicating a shift in pattern score between samples (i.e. change in the average CHH enzyme methylating the element). **g** Schematic representation of the methylation patterns of CMT2 and DRM2 identified in this study: CMT2-mediated methylation has higher stochasticity, possibly reflecting lower processivity, and a unimodal distribution among cells; DRM2-mediated methylation has lower stochasticity, possibly reflecting higher processivity, and has a bimodal distribution among cells

## SRBrowse, a tool for visualising and analysing BS-seq data at single-read resolution

The most popular genome browsers, including UCSC genome browser and Integrative Genomics Viewer (IGV) [42], are limited in their ability to load high-resolution data on-the-fly without creating a large memory footprint, significantly increasing the load times of browser displays or requiring pre-processing of data. In order to visualise single read data, it is necessary to convert aligned read data (e.g. SAM/BAM files) into track files (such as GFF) or compressed indexed files (BED, TDF, etc.) suitable for fast retrieval.

In order to make browsing and analysing BS-seq data at single-read resolution more accessible, we developed a genome browser specifically designed for visualising BS-seq data at a single-read level, which we called SRBrowse. SRBrowse is web browser-based and can run on a local computer or server with minimal software requirements (see the repository README file on installation). Importantly, SRBrowse allows users to load data into the browser view and monitor its alignment progress from the same interface. Typical

steps for loading and displaying data appear in Additional file 1: Figure S8. All data loaded through SRBrowse is aligned using bowtie2 and stored in indexed files allowing optimised access to the read data.

## Discussion

We presented a novel analysis pipeline for extracting additional layers of information from NGS BS-seq data. The pipeline uses data of individual read methylation states and the distribution of DNA methylation within reads to identify patterns that augment information regarding the averaged methylation signal. Using this pipeline, we were able to define characteristic features of reads and regions methylated by two non-CG-methylating enzymes, CMT2 and DRM2, in *A. thaliana*. Specifically, we found that *A. thaliana* mutants of CMT2 and DRM2 present stereotypical CHH methylation patterns that are robust to background methylation and consistent among different mutated alleles and species (Fig. 5, Additional file 1: Fig. S4b-c). These patterns are also independent of demethylase activity: in the absence of demethylase activity, the same distinct patterns are observed in regions regulated by each enzyme (Fig. 2c, d, Additional file 1: Fig. S3). On the one hand, *cmt2* mutants have mainly highly methylated reads, and methylation is concentrated within specific regions; on the other hand, *drm2* and *drd1* mutants have mainly lowly methylated reads, and methylation is distributed stochastically within and among reads. In other words, our analysis suggests that, compared to DRM2, CMT2-methylated regions presents more stochastic variation of methylation level among cells. In contrast, DRM2-methylated regions present distinct subpopulations of methylation states, with less stochastic variation (Fig. 6g).

By analysing methylation patterns at single-read resolution, where each read bears the characteristics of the methylation mechanism in a single genome (i.e. of the same DNA molecule), our data can make predictions regarding the enzymatic activity of methylases. The assumption that the identified patterns relate to enzyme activity is strengthened by our results, which suggest that the distinct methylation patterns of DRM2 and CMT2 are not influenced by demethylation activity (Fig. 2), nor correlated to chromatin structure (Fig. 4). Accordingly, we predict that differences in methylation stochasticity reflect a distinction in the processivity of the methylases, specifically, that DRM2 has higher CHH methylation processivity than CMT2.

Variation in DRM2 and CMT2 methylation characteristics could relate to the distinct genomic targets of these enzymes. DRM2 methylates mostly short euchromatic-TE sequences located next to genes and CMT2 methylates mainly long heterochromatic-TE sequences [13, 14]. Thus, the bimodal distribution of DRM2-methylated read subpopulations, in terms of methylation level, could relate to the ability of DRM2 methylation to regulate genes within particular cell types or under certain conditions (Fig. 6g), such as in the formation of lateral root development [43]. In contrast, the CMT2 methylation pattern, which is low but uniform, correlates with constant need to silence heterochromatic TEs (Fig. 6g).

Based on the variation of these patterns between CHH-methylating mutants, we designed a classifier that scores short regions of 30 base pairs and collections of regions within functional elements (such as genes, exons or TEs). This score provides an arbitrary scale to differentiate between DRM2-like and CMT2-like CHH methylation patterns. The comparison among species highlights the ability of the classifier to predict the presence

or absence of CMT2/DRM2 in species for which mutants have not yet been developed, such as *M. polymorpha*. The classifier is robust to differences in sample heterogeneity, and is able to differentiate between methylation patterns even within highly specific tissue samples (Fig. 6).

While DRM2 in plants as well as human DNMT3 are monophyletic and distantly related to CMT2, DRM2 is the only enzyme that has a rearranged catalytic domain [16]. Therefore, our findings that DNMT3 and CMT2 have similar CHH methylation characteristics suggest that different DNMTs can have similar methylation mechanisms, and substantiates the hypothesis that DNMT3 CHH methylation activity in early land plants has been replaced by CMT2 in angiosperm [16]. Moreover, the unique, highly processive methylation activity we predicted for DRM2 could be associated to its exclusive rearranged catalytic domain rather than to its general homology to DNMT3 enzymes.

## Conclusion

Overall, the analyses of methylation profiles we present here demonstrate the potential of studying patterns of variation in BS-seq data through single-read analyses, which provide new biological insights on the writing, erasing, and readout mechanisms of CHH methylation. The tool we developed can facilitate further studies of methylomes at single-read resolution.

## Methods

### BS-seq alignment

All code used for the read analysis pipeline is deposited in a public software repository (https://github.com/zemachlab/srbrowse) under a CC-BY-4.0 License. For aligning reads from BS-seq data, we used bowtie2 with a Node.js-based wrapper. The method we used for aligning BS-seq data is based on a previously described pipeline [14]. The wrapper converts the reference assembly to C-to-T and G-to-A sequences before bowtie2 indexing; each strand is converted manually so that each genome index consists both of forward and reverse strand versions of each scaffold. BS-seq reads are converted either C-to-T or G-to-A depending on whether the read is a left or right mate (in the case of paired ends reads), with the original read data stored for collecting methylation information after alignment. Bowtie2 was run with the end-to-end search algorithm. For all datasets, a minimum score of 0 was used (i.e. no mismatches or gaps). Reads that mapped to more than one position were discarded. Aligned reads were then sorted and exact duplicates removed.

### Analyses of BS-seq data

Analyses consist of three stages: (1) identifying short regions according to the region selection parameters (see the "Designing a single-read analysis pipeline for CHH methylation" section), (2) extracting reads overlapping with each region from the selected samples and (3) averaging read data. Region selection parameters were optimised to ensure sufficient data for low coverage samples (Additional file 1: Fig. S1a-c). Functional elements are first selected according to an annotation provided in GFF format. Next, sites of specific methylation contexts are identified based on the reference sequence of the element (e.g. CHH sites), per strand. Separation to strands is important for asymmetrical contexts such as CHH. Regions are defined by iterating through sites until the the number of required

CHH sites within the defined region size is reached. Reads from aligned BS-seq samples are retrieved from indexed lists of reads and stored as binary arrays where each CHH site is represented as either unmethylated (0) or methylated (1).

The read methylation data of a specific region were analysed to produce methylation features (Fig. 1b) of individual reads and their associated regions. For individual reads, these features are (1) read methylation, the mean methylation of CHH sites within the read, and (2) read stochasticity, the number of changes between methylation states between adjacent sites (for illustration, see Fig. 1b). Importantly, features of individual reads do not refer to the overall methylation of the read, but only to methylation at the sites included in the specific region. For regions, the methylation features are (1) mean read methylation, (2) standard deviation of read methylation, (3) mean read stochasticity and (4) standard deviation of site methylation. The last feature is not based on single-read data but rather the averaged methylation signal at each site.

The output data of this analysis can be either at the level of individual reads, individual regions or whole functional elements. For reads data, the output is an array of reads for each sample from all regions matching the selection parameters. For regions, the output is an array of regions with features derived from averaged read data as explained above. The regions also have positional data relating to their parent element, length, and any other genomic features of interest (e.g. GC content). For whole functional elements, the output is an array of elements such as exons or transposable elements, where methylation features relate to the average of all regions identified with in the functional element. The exclusion of regions based on methylation or coverage, prior to averaging whole elements, is important to reduce background of unmethylated regions. Unless stated otherwise, regions were selected with a minimum of 10% methylation average and 4 overlapping reads. Functional elements that contained at least two such regions were selected. While increasing the minimum regions per element improves coverage per element, it can bias the analysis to longer elements.

### Statistical analyses

All statistical analyses we performed on either read, region or element data resulting from the above pipeline, using Python 3.6 along with the following libraries: matplotlib, numpy, scipy, statsmodels, pandas and seaborn. K-density plots were produced using seaborn.distplot (which uses statsmodels.nonparametric.kde.KDEUnivariate) with Gaussian kernel shape and Scott's Rule of Thumb bandwidth. For ANOVA of methylation features, we used methylated regions from the CHH methylation mutants *drd1* and *cmt2* (composed of data from *cmt2-4*, *cmt2-5* and *cmt2-6* mutant alleles). Methylation features of the regions were provided as independent variables, and sample source (0 for *drd1*, 1 for *cmt2*) as the dependent variable. The results of the ordinary least squares model are presented in Additional file 1: Table S1. For the classifier, the resulting coefficients of the independent variables were scaled so that pattern score is defined between 0 and 1. Linear regression for scatter plots were conducted using scipy.stats.linregress.

### Data sources

The following assemblies were used for aligning reads: GCF_000001735.4 (*A. thaliana*), GCF_000002425.4 (*P. patens*), *O. sativa* v7.0, GCF_000188115.4 (*S. lycopersicum*), *M. polymorpha* v3.0, GCF_000001405.39 (*Homo sapiens*). The following annotations were

used for genes and TEs: Araport 11 TE annotation from TAIR [44] for *A. thaliana*; *P. patens* TE annotation was downloaded from CoGe, and information from a *P. patens* Repeatmasked assembly (v3.3) was downloaded from Phytozome to increase the resolution of LTR-TEs families; TEs were annotated de novo for *M. polymorpha* using REPET v3.0 [45, 46]; Repeatmasked assemblies were downloaded from Phytozome for *O. sativa* (323) [47] and *S. lycopersicum* (ITAG 3.2) [48]; GCA_000001405.28 gene annotation [49] was used for *H. sapiens.*

Whole genome BS-seq data from the following studies was used (for a full list of accessions see Additional file 2): GSE41302 for *A. thaliana cmt2, drm2, drd1* mutants [14], GSE64569 for *A. thaliana ros1* mutants [50], GSE33071 for *A. thaliana rdd* triple mutant [51], GSE38935 for *A. thaliana dme* mutants [52], GSE87170 for *A. thaliana* sperm and vegetative nucleus wild type and *cmt2* and *drm1/2* mutants [31], GSE79746 for *A. thaliana drm2* and *cmt2* mutants [53], GSE39901 for *A. thaliana cmt2* mutant [30], GSE43857 for *A. thaliana* ecotypes Gro-3, Kz-9 and Neo-6 [54], PRJNA350766 and GSE118153 for *P. patens* wild type samples [16, 55], SRP101412 for *M. polymorpha* wild type (thallus) samples [40], GSE81436 for *O. sativa* wild type sample [38], GSE108527 for *O. sativa drm2 ddm1* mutant [56], SRP008329 for *S. lycopersicum* wild type sample [57], SRP081115 for *S. lycopersicum slnrpd1* mutant [35], and GSE47966 for *H. sapiens* frontal cortex samples [39].

## Supplementary information

---

**Additional file 1:** Supplementary figures.

**Additional file 2:** List of NCBI Single-Read Archive (SRA) accessions used in this study.

**Additional file 3:** Review history.

---

**Peer review information**

**Acknowledgements**
We would like to thank Ohad Roth for contributing to an early draft of the manuscript. We would also like to thank all members of the Zemach lab and Nir Ohad's lab for their critical feedback on the study, and the reviewers for their important feedback on our manuscript.

**Review history**
The review history is available as Additional file 3.

**Authors' contributions**
AZ conceived the study. AZ and KDH designed the single-read analysis pipeline. KDH designed and wrote the software implementing the pipeline and analysed data. AZ and KDH co-authored the manuscript. The authors read and approved the final manuscript.

**Funding**
This work was supported by the European Research Council (ERC, 679551) and Israel Science Foundation (1636/15) to A.Z.

**Availability of data and materials**
The source code of the software presented is available on GitHub [58]. The version of the code used for the analyses in this paper is available on Zenodo [59]. The code is released under a CC-BY-4.0 license. All data used in this study is publicly available, as indicated in the "Methods" section. A comprehensive list of NGS accessions used in this study is available in Additional file 2.

**Ethics approval and consent to participate**
Not applicable.

**Consent for publication**
Not applicable.

**Competing interests**
The authors declare that they have no competing interests.

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

## 