## [**Additional file 3** Review history. · Genome Biology]

Review History

First round of review

Reviewer 1

Were you able to assess all statistics in the manuscript, including the appropriateness of statistical tests used?

No.

Were you able to directly test the methods? No.

Comments to author:

Harris and Zemach present a method for analyzing whole-genome bisulfite sequencing data on an individual read basis and classifying CHH methylated reads based on their probable enzymatic origin. This is a rather interesting approach. Biologically, the key detail this paper gives is that at a cellular level, CMT2 primarily has greater among site variation, while likely being active in a broader swath of cells, while DRM2 has lower among site variation and appears more restricted in specific cells. This tool will be of use in addressing a number of potential questions, however, its overall applicability is very limited and primarily to those specialists working in plant DNA methylation, outside of plants, this method will not be of much use. I think the overall approach is largely sound.

1) Line 53, it looks like there may be an error in converting the text: "TEs containing 5 CHH sites with $\zeta=5\%$ methylation"

2) While I understand the author's rationale for limiting their analyses to regions with 5 CHH sites per 30bp, I am still worried that this will exclude a large number of regions. It would be useful to know how much of the genomes studied are excluded based on this criteria and what proportion of CHH methylation in the genome is simply ignored.

Reviewer 2

Were you able to assess all statistics in the manuscript, including the appropriateness of statistical tests used?

No.

Were you able to directly test the methods? No.

Comments to author:

The authors present an interesting manuscript, which promises to get more information out of the wealth of DNA methylome data that has been generated. Their approach also aims to better interrogate between-cell variation in methylomes within heterogenous samples. There is certainly merit to this research and I found the manuscript interesting.

Briefly, the authors propose an alternative analysis method which, rather than averaging methylated cytosine calls across all reads (as is standard practice), considers the methylation status of individual reads. This is a very interesting idea. There is plenty of evidence that methylation varies between cells, but we have all largely been content to work from average signals across populations of cells. The authors demonstrate that their approach works using a range of data from Arabidopsis, comparing wild-type and DNA methylation machinery mutant lines. They make a convincing case that analysis by three metrics at single-read level quantifies differences between the methyltransferases CMT2 and DRM2; their approach allows them to pinpoint differences in processivity of these two enzymes, as shown by the differing patterns of methylation in regions targeted by each enzyme. Further analysis of mutant DNA methylomes determines that these patterns are not driven by the competing activities of demethylase enzymes. The authors next develop a classifier for individual read methylome data, which allows each read to be attributed to either a CMT2-like or DRM2-like enzyme. The utility of this is that data can now be assessed from species for which much less biochemical characterisation of methyltransferases has been conducted. The authors assess four other plant species and identify that some appear to have both DRM2 and CMT2-like activity, whilst others have only CMT2-like activity. This result fits with what is already known of the enzymes encoded by the genomes of each species. Further, they apply the classifier to mammalian neuronal methylome data, predicting a single CMT2-like enzyme is responsible for neuronal non-CG methylation. In these tests the classifier performs well,

demonstrating cross-species applicability. Next, the authors examine tissue-specific methylome data from Arabidopsis, determining that the relative activity of DRM2 and CMT2 varies between tissues; again, this fits with observations from prior studies of the activity of these enzymes.

In summary I think these tools will be of use to other researchers of DNA methylation across multiple species. They allow us to gain substantially more insight into biological processes, and make meaningful predictions of biochemical activity, than current approaches. I am not aware of any comparable tool available currently. The approach also contributes to the growing interest in cell-specific analyses. I expect the tools to be well adopted and the manuscript to consequently be well-cited. I am in favour of publication. Below are some minor changes I suggest.

- Fig2a needs a bit more description in the legend. As it stands, I don't fully understand it. If the panels report all reads containing 5 Cs and with a minimum of 1 methylated C, how can an average region methylation level of 5% exist? Is this the average of the region rather than the read? If so that should be stated clearly in the legend. The absence of reads with 0 methylated Cs from this analysis means that is not intuitive.

- It would help the reader if the relevant functional differences (p.4 ln.9-14) were also labelled directly on Fig. 1b.

- Fig 2b is somewhat confusing. In the main text related to this section (p.4 ln.31-33) the authors state analyses were conducted on 30bp regions with max 5 CHH sites. But Fig. 2b appears to be working on a different set of regions, with up to 15 CHH sites. Fig 2c then presumably returns to the original set of regions, since it only reports up to 5 Cs, but this is not explicitly stated. Is one set of regions mutually exclusive of the others, or are they a completely different segmentation of the genome?

- The x axis in both panels a and b of Fig 2 is read methylation, but expressed using different scales. These are not, but should be, explained clearly in the legend. Also, acronyms should be defined - I am assuming that Cs refer to methylated CHH sites, but this should be stated.

- p.5 ln.43-46. The conclusion that demethylases reduce methylation entirely in specific cells is overly strong. Phrased this way it implies that all methylation is removed from all DNA in specific cells. The data don't support this; they indicate the demethylases reduce methylation to zero in some reads, but they don't tell us whether or not this 0 methylation reads all come from the same cell(s).

- p.9 ln.20. State what drm12 is a mutant of.

Reviewer 3

Were you able to assess all statistics in the manuscript, including the appropriateness of statistical tests used?

No.

Were you able to directly test the methods? No.

Comments to author:

I really like this paper, which is trying to disentangle mechanisms of methylation by look at the pattern of methylation polymorphism in pools of cells — i.e., normal bsSeq data. There is literature on doing this in humans you may want to cite — check, e.g., Eran Halperin's work. But of course it won't be directly relevant given that the RdDM pathway does not exist in human.

What you are doing clearly works, and your approach deserves to be published. I'm not quite sure your interpretations are right: in particular, I am not convinced that what you are seeing is strictly due to enzymatic properties. I'm not an expert, but I can imagine some of my more biochemical colleagues being upset by such a claim. You have no direct proof, and you have certainly not ruled out alternatives. Also, the correlation with H3K9 in Figure 4C looks quite strong. I would strongly suggesting toning down this language; not call it "enzyme score", etc. Your results are interesting enough without it. Fine to speculate in the Discussion, of course.

Analogously, I would not use "processivity" to describe a statistical pattern. Not only is it confusing to non-biochemists, it is also misleading as it strongly implies an explanation for the pattern that you have not demonstrated. Again: stick to the data. Fine to discuss interpretations in the Discussion. The paper will be easier to read, you will not be wrong, you will annoy fewer people, and you will get your message across more efficiently.

Question: how do these pattern relate to CHG?

Minor points:

p. 5, l 31 "which their activity" — something is missing here

p. 5, l 53 "?=5%" — strange character

Dear Editor

We would like to thank the reviewers for bringing our attention to several issues that could affect the clarity of our results and the precision of our arguments. We have carefully considered their comments and integrated their suggestions. We have also revised any other errors we found in the manuscript. Below we provide a point-by-point response to all of the reviewers' comments.

Reviewer #1:

1) Line 53, it looks like there may be an error in converting the text: "TEs containing 5 CHH sites with \geq 5% methylation"

Answer:

Line 53 has been fixed.

2) While I understand the author's rationale for limiting their analyses to regions with 5 CHH sites per 30bp, I am still worried that this will exclude a large number of regions. It would be useful to know how much of the genomes studied are excluded based on this criteria and what proportion of CHH methylation in the genome is simply ignored.

Answer:

As we were aware that the selection of regions containing 5 CHH sites of a maximum length of 30 base pairs potentially limits the analysis to a limited subset of regions and may not be representative of the majority of CHH-methylated regions, we plotted the distribution of the length of regions containing 5 CHH sites in a wild type *A. thaliana* sample (Figure S1c). This shows that the subset we used (\leq 30 bp) contains 58% of regions.

Reviewer #2:

1. Fig2a needs a bit more description in the legend. As it stands, I don't fully understand it. If the panels report all reads containing 5 Cs and with a minimum of 1 methylated C, how can an average region methylation level of 5% exist? Is this the average of the region rather than the read? If so that should be stated clearly in the legend. The absence of reads with 0 methylated Cs from this analysis means that is not intuitive.

Answer:

We have added the 0 Cs column to Figure 2a. It was previously excluded to indicate the pattern seen specifically among methylated reads. In this analysis, reads are selected from regions with an average of at least 5% CHH methylation. We have edited the legend of the figure to make this clear.

2. It would help the reader if the relevant functional differences (p.4 ln.9-14) were also labelled directly on Fig. 1b.

Answer:

As the functional differences between regions are interpretations of the results, we preferred not to present them in the initial figure which establishes the means with which regions were analysed. We were also encouraged by the other reviewers to avoid attributing the identified patterns to functional aspects of the enzymes prior to the discussion. At this point in the main text we preferred only to allude to the possible biological interpretations of these features.

3. Fig 2b is somewhat confusing. In the main text related to this section (p.4 ln.31-33) the authors state analyses were conducted on 30bp regions with max 5 CHH sites. But Fig. 2b appears to be working on a different set of regions, with up to 15 CHH sites. Fig 2c then presumably returns to the original set of regions, since it only reports up to 5 Cs, but this is not explicitly stated. Is one set of regions mutually exclusive of the others, or are they a completely different segmentation of the genome?

Answer:

Figure 2b is a different type of analysis that identifies the properties of individual reads regardless of the originating region, and uses different region selection parameters than Figure 2a. This analysis shows how individual reads can have different properties depending on the methylating enzyme, and how these properties are correlated differently in each mutant. Region selection was not clearly indicated in the text. In this panel, three different sets of regions are used, with 5, 10 or 15 CHH sites. Panels c-d are the same as panel a in terms of the type of analysis. We have indicated how the regions were selected in the figure legend.

4. The x axis in both panels a and b of Fig 2 is read methylation, but expressed using different scales. These are not, but should be, explained clearly in the legend. Also, acronyms should be defined - I am assuming that Cs refer to methylated CHH sites, but this should be stated.

Answer:

We have relabelled the x-axes on panels a and c-d of Figure 2 as "Methylated CHH sites" to clearly distinguish them from Figure 2b and to make it clear what "Cs" refers to. We have also added an explanation to the legend.

5. p.5 ln.43-46. The conclusion that demethylases reduce methylation entirely in specific cells is overly strong. Phrased this way it implies that all methylation is removed from all DNA in specific cells. The data don't support this; they indicate the demethylases reduce methylation to zero in some reads, but they don't tell us whether or not this 0 methylation reads all come from the same cell(s).

Answer:

The intention was that, in specific cells, entire regions appear to be demethylated. As it was phrased, it appeared to suggest that the statement applied to methylation across multiple regions in the same cells. As this interpretation of this result is also not crucial for our main conclusions, we chose to remove it.

6. p.9 ln.20. State what drm12 is a mutant of.

Answer:

We have indicated specifically what drm12 is a mutant of.

Reviewer #3:

1. I really like this paper, which is trying to disentangle mechanisms of methylation by look at the pattern of methylation polymorphism in pools of cells — i.e., normal bsSeq data. There is literature on doing this in humans you may want to cite — check, e.g., Eran Halperin's work. But of course it won't be directly relevant given that the RdDM pathway does not exist in human.

Answer:

We have cited Eran Halperin's work in the introduction. It bears some similarities to the Huan et al. 2018 paper we cited (both use CG methylation data, and treat individual reads as representing distinct methylation states).

2. What you are doing clearly works, and your approach deserves to be published. I'm not quite sure your interpretations are right: in particular, I am not convinced that what you are seeing is strictly due to enzymatic properties. I'm not an expert, but I can imagine some of my more biochemical colleagues being upset by such a claim. You have no direct proof, and you have certainly not ruled out alternatives. Also, the correlation with H3K9 in Figure 4C looks quite strong. I would strongly suggesting toning down this language; not call it "enzyme score", etc. Your results are interesting enough without it. Fine to speculate in the Discussion, of course.

Answer:

Hypothetically, if the pattern we identified is not enzyme-specific, then it might be possible to identify regions methylated by one enzyme that showed the pattern of the other enzyme. We would also expect to find variation in the pattern dependent on genomic features such as different types of elements (genes, transposable elements, intergenic regions) and chromatin structure. To this end, we checked a number of other factors that could have created the distinct patterns we identified in CMT2- and DRM2-methylated regions: GC content, histone markers, and demethylase activity. As expected, we found that GC content was higher, the H3K4me3 signal was lower and the H3K9me2 signal was higher in regions methylated by CMT2. This agreed with previous results (Zemach et al. 2013). However, we did not find a correlation between GC content and histone markers and the patterns we observed; specifically, methylated regions from the CHH methylase mutants behaved the same (produced the same "enzyme score") regardless of their GC content and histone marker signals (Figure 4). Linear regressions of this scatter plot data suggest that GC content and histone markers do not explain differences in our "enzyme score" feature.

Thus, while we understand that this is not direct evidence of the enzymatic mechanism that contributes to these patterns, we conducted the analyses mentioned above specifically to

question our hypothesis. In addition, the label “enzyme score” referred to the association of a certain pattern with each enzyme. Nonetheless, for the sake of avoiding confusion or unnecessary assumptions about the significance of the patterns we identified, we have renamed “enzyme score” to “pattern score”. We have also avoided referring to the patterns we identified as relating directly to enzymatic activity, apart from in the Discussion, where we offer this as a prediction.

3. Analogously, I would not use "processivity" to describe a statistical pattern. Not only is it confusing to non-biochemists, it is also misleading as it strongly implies an explanation for the pattern that you have not demonstrated. Again: stick to the data. Fine to discuss interpretations in the Discussion. The paper will be easier to read, you will not be wrong, you will annoy fewer people, and you will get your message across more efficiently.

Answer:

As above, to avoid unnecessary assumptions regarding the mechanistic interpretation of our results, we have avoided use of the term “processivity” in describing our results, instead using the term “stochasticity” which had already been used to describe the distribution of methylation within reads. We also changed the title, using the term “contiguous” to describe the observed pattern rather than arguing that we have identified the difference in the enzymatic mechanism. Any mention of processivity has been kept to the Discussion.

4. Question: how do these pattern relate to CHG?

Answer:

Similarly to CHH methylation, CHG methylation in *A. thaliana* also shows “partial” methylation, indicating differential regulation of methylation among cells. It is possible to adapt the current pipeline to analyse CHG methylation, to determine whether there are different patterns also, and to correlate these patterns to genomic features. We chose to focus on CHH methylation given the density of CHH sites, which yields more regions that suit our single-read analysis pipeline, but also due to the fact that CHH sites are methylated by two different methylases.

5. p. 5, l 31 "which their activity" — something is missing here

Answer:

Fixed, typo of “the activity of which”.

6. p. 5, l 53 "?=5%" — strange character

Answer:

Fixed this character.

Yours sincerely

The authors

Second round of review

Reviewer 2

This is a nice paper. The authors have addressed my comments appropriately and I'm now in favor of accepting the manuscript.

I did not have time to run their code myself, but did look at the GitHub repository briefly. It appears everything is present that was used in the analysis. I do have one final request for the authors; the readme document is brief and likely not adequate for the average user to run your code without some trial and error. Easy reuse by others = happy colleagues and citations for you, so please consider adding more detail. I do not think this is a reason to delay accepting the manuscript though - the authors can make this update to GitHub very easily and quickly.